# Clinical Application of Metagenomic Next-Generation Sequencing in Patients with Hematologic Malignancies Suffering from Sepsis

**DOI:** 10.3390/microorganisms9112309

**Published:** 2021-11-06

**Authors:** Wang-Da Liu, Ting-Yu Yen, Po-Yo Liu, Un-In Wu, Prerana Bhan, Yu-Chi Li, Chih-Hung Chi, Wang-Huei Sheng

**Affiliations:** 1Department of Internal Medicine, National Taiwan University Hospital and National Taiwan University College of Medicine, Taipei 100, Taiwan; b95401043@ntu.edu.tw (W.-D.L.); poyu.liu@gmail.com (P.-Y.L.); uninwu@gmail.com (U.-I.W.); 2Department of Medicine, National Taiwan University Cancer Center, Taipei 106, Taiwan; 3Department of Pediatrics, National Taiwan University Children’s Hospital, Taipei 100, Taiwan; ytingyu@gmail.com; 4Department of Pathobiology and Population Sciences, Royal Veterinary College, University of London, London AL9 7TA, UK; 5LIHPAO Life Science Cooperation, New Taipei City 251, Taiwan; prerana.bhan@libobio.com (P.B.); ryan.li@libobio.com (Y.-C.L.); brian.chi@libobio.com (C.-H.C.); 6Department of Medical Education, National Taiwan University Hospital, Taipei 100, Taiwan; 7School of Medicine, National Taiwan University College of Medicine, Taipei 100, Taiwan

**Keywords:** blood culture, neutropenia, septic shock, polymerase chain reaction

## Abstract

Background: Sepsis remains a common but fatal complication among patients with immune suppression. We aimed to investigate the performance of metagenomic next-generation sequencing (mNGS) compared with standard microbiological diagnostics in patients with hematologic malignancies. Methods: We performed a prospective study from June 2019 to December 2019. Adult patients with hematologic malignancies and a clinical diagnosis of sepsis were enrolled. Conventional diagnostic methods included blood cultures, serum galactomannan for Aspergillus, cryptococcal antigen and cytomegalovirus (CMV) viral loads. Blood samples for mNGS were collected within 24 h after hypotension developed. Results: Of 24 patients enrolled, mNGS and conventional diagnostic methods (blood cultures, serology testing and virus RT-PCR) reached comparable positive results in 9 cases. Of ten patients, mNGS was able to identify additional pathogens compared with conventional methods; most of the pathogens were virus. Conclusion: Our results show that mNGS may serve as adjunctive diagnostic tool for the identification of pathogens of hematologic patients with clinically sepsis.

## 1. Introduction

Septic shock in patients with hematologic malignancies are common and could carry high risk for mortality [1]. Besides, the situation of febrile neutropenia, the use of corticosteroids, immunosuppressive drugs on T-cell function, and newly developed targeted molecular therapy has led to more complicated scenarios, starring a range of pathogens, such as viruses, bacteria, fungi, and parasites [2,3,4]. Prolonged broad-spectrum antibiotics administration might be considered a risk of selection of multidrug-resistant bacteria, while early antimicrobial de-escalation has a significant impact on the ecology of flora, which relies on pathogen identification [5,6]. However, the limitations of conventional culture methodologies have challenged physicians to make a precise diagnosis and de-escalate antimicrobial agents appropriately. Culture-independent methods, such as the nucleic acid amplification test, might help towards an adequate diagnosis.

Metagenomic next-generation sequencing (mNGS) provides a sensitive and thorough approach for detecting all pathogens in clinical samples, ranging from conventional bacteria to atypical and rare pathogens [7,8,9]. The advantage of rapid diagnosis is believed to bring extra benefit among patients with hematologic malignancies. Previous studies mainly focused on detecting pathogens from respiratory specimen and cerebrospinal fluid [10,11,12,13], while clinical application among patients with bloodstream infections among patients with hematologic malignancies remains less discussed. A previous investigation by Gyarmati et al. which focused on nine patients with febrile neutropenia demonstrated metagenomic analysis is useful for pathogen identification, including bacterial, fungal, and viral infection [14].

In this study, we aimed to expand the clinical application of mNGS in pathogen identification, and compare the results between mNGS and conventional methods, among hematologic patients with the clinical diagnosis of sepsis.

## 2. Materials and Methods

### 2.1. Study Population

From June 2019 to December 2019, we conducted a prospected study at National Taiwan University Hospital, which is a tertiary medical center with 2400-beds, located in northern Taiwan. Patients aged 20 years or older with hematologic malignancies and a clinical diagnosis of sepsis were screened. The diagnosis of sepsis was based on the quick sequential organ failure assessment (qSOFA) score from the 2016 international consensus for sepsis and septic shock [15]. Patients with at least two of the following clinical criteria suggestive of septic shock, including a respiratory rate of 22/min or greater, altered mental status, or systolic blood pressure of 100 mmHg or less., were enrolled. Infectious disease physicians were consulted to visit the enrolled patients with an aim to exclude non-infectious processes, such as congestive heart failure, hypovolemia, and rheumatic or endocrine disorders. Besides, only those who were documented with a body temperature of more than 38 degrees Celsius and an episode of systolic pressure below 100 mmHg were included in order to strengthen the diagnostic accuracy of sepsis. Blood tests, including two bacterial cultures and one fungal culture, were arranged among every enrolled patient. Blood test for mNGS was performed within 24 h after the onset of a hypotension episode. In addition, serum galactomannan test, cryptococcal lateral flow antigen test and cytomegalovirus (CMV) reverse transcription-polymerase chain reaction (RT-PCR), and sputum and urine bacterial culture were performed for all enrolled patients upon hospitalization. For patients with respiratory symptoms or pulmonary infiltrates found via chest radiograph, sputum fungal culture, sputum acid-fast stain with mycobacterial culture were administered, along with sputum RT-PCR tests for pneumocystis and CMV. Patients with diarrhea were assigned to have stool culture for *Campylobacter sp*., *Salmonella sp.*, *Shigella sp*. and *Clostridium difficile*. A test for the Clostridium difficile toxin RT-PCR was also performed. The study has been approved by the Institutional Review Board (Ethics Committee) of National Taiwan University Hospital (IRB number, 201902079RIND, approval date: 8 May 2019).

### 2.2. Next-Generation Sequencing Methodology

About 5–10 mL of fresh blood was drawn from each subject and collected into Streck vacutainer tubes. These tubes were stored at ambient temperatures before plasma separation. Collection tubes were centrifuged at 1600× *g* for 10 min at 4OC to separate the plasma. One mL of plasma was then transferred to DNA LoBind and centrifuged at 16,000× *g* for 10 min at 4OC to reduce the amount of host WBC, the gDNA of which might interfere with the later process. Next, cfDNA was extracted from 200 μL plasma utilizing TIANamp Micro DNA Kit (DP316, TIANGEN BIOTECH, Beijing, China) following the manufacturer’s operational protocol. Subsequently, the DNA libraries were constructed through consecutive reactions that include end-repair, adaptor ligation, PCR amplification and purification. The DNA library quality was assessed using Qubit high sensitivity DNA assay and capillary electrophoresis (BioAnalyzer DNA 1000 analysis kit). The qualified libraries were further subjected to DNA circularization using MGIEasy circularization kit, followed by DNA nanoball (DNB) formation using the rolling circle replication mechanism. Finally, upon DNB formation the sequencing was performed on a MGISEQ-200 sequencer (MGI Tech Co. Ltd., Shenzhen, China), resulting in greater than equal to 300 million 100-base single end reads.

### 2.3. Bioinformatics

Upon generation of the sequencing result, the quality control of the raw data was conducted by trimming the adaptor sequences, low quality reads (Phred score <20 and reads lower than 35 bp were discarded) and the reads with high content of N bases using the SOAPnuke program. Next, the host reads were eliminated after mapping the sequence to human reference sequence hg1,9 using the Scalable Nucleotide Alignment Program (SNAP). Subsequently, the rRNA contamination was also removed. Clean reads were obtained after quality control of raw data and the base number was calculated using the formula “read number * read length”. Furthermore, the pathogen sequences were mapped to NCBI databases. The results indicate the top ten species based on the sample type, such as bacterial, virus, fungi, etc. Information, such as read number and total classified reads, will also be listed along with the % of relative abundance. The identification workflow includes: (1) identification of species using the Kraken2 software (version 2.0.6-beta) with default parameters and the accompanying database (version Kraken_dbk35_all). (2) elimination of the false positive data and (3) calculation of the relative abundance. All these steps are used to systematically classify the pathogens using reference databases (MGI in-house database and RefSeq NCBI databases [NCBI database retrieve date: 1 August 2021]) for bacteria, virus, fungi, and protozoa, respectively. Lastly, the pathogen fast identification report was generated using the MGI ZLIMS automated platform. The procedures of mNGS quality control referenced the guidelines set by Kunin and Zhou [16,17].

## 3. Results

Demographic features of the patients in the current study are provided in Table 1. Distribution and classification of high-quality reads among the blood samples via mNGS are presented in Table 2. A total of 24 hematologic patients were enrolled. Nine (37.5%) patients were diagnosed as having acute myeloid leukemia, eight (33.3%) with diffused large B-cell lymphoma, four (16.7%) with multiple myeloma, and one (4.2%) with acute lymphocytic leukemia, T-cell large granular lymphocytic leukemia and mantle cell lymphoma, respectively (Figure 1). The majority of patients were diagnosed with pneumonia (9/24 [37.5%]), followed by intraabdominal infection (7/24 [29.2%]), bloodstream infection (3/24 [12.5%]) and urinary tract infection (2/24 [8.4%]) (Figure 2). Fourteen (58.3%) patients were diagnosed with neutropenia (absolute white blood cell < 500 cells/mm^3^) when sepsis developed.

In our results, mNGS and conventional methods (blood cultures, serology testing and virus RT-PCR) were both positive in 14 (58.3%) cases and were both negative in four (16.7%) cases. Four patients were positive by mNGS only (16.7%) and two were positive by conventional methods only (8.3%) (Figure 3). For the double-positive subgroup, a high proportion of complete matching (5/14, 35.7%) and partial matching (at least one pathogen identified in the test was confirmed by the other) (7/14, 50%) was seen, with only two conflicts between mNGS and conventional methods.

Our study showed that mNGS and conventional diagnostic methods reached comparable positive results in nine cases (case no. 1 to no. 9). Among five patients with positive findings (case no. 1 to no. 5), three are AML patients with bacteremia, while the other two patients had clinical diagnosis of interstitial pneumonitis owing to CMV and *P. jirovecii*. In ten patients (case no. 10 to no. 19), mNGS was able to identify additional pathogens from the blood samples compared with conventional methods. Aside from bacteria, the extra information provided by mNGS was mainly virus, including CMV, human mastadenovirus C, and herpes simplex virus type 1, which was discovered from seven patients.

For the 15 patients who obtain equal or more microbiologic evidence via mNGS (case no. 1 to no. 5 and no. 10 to no. 19), eight (53.3%) patients had blood sampled prior to effective antimicrobial agents given and four (26.7%) patients had blood sampled on the same day when fever flared-up. For three patients, more information was required by conventional diagnostic methods (case no. 20 to no. 22). The blood cultures yield *K. pneumoniae*, *A. veronii* and *E. coli*, while mGNS showed negative results. All the three patients received mNGS samplings 1 to 2 days after empirical antimicrobial therapy. Discrepancies between the two diagnostic methods were found in two patients (case no. 23 and no. 24). Both patients received mNGS samplings two days after empirical antimicrobial therapy.

## 4. Discussion

Our study successfully demonstrated the advantage of mNGS in diagnosing pathogens via serum samples among patients with hematologic malignancies who developed sepsis. Previous studies have shown the superiority of NGS methods targeting the 16S ribosomal RNA gene in the diagnosis of bacterial infection among immunocompromised patients [18,19,20]. Metagenomic NGS further provides effective diagnostic approaches in diagnosing atypical pathogens such as virus, fungi, and mycobacterium [7,14], which are more commonly seen among patients with hematologic malignancies than the general population. Our study also demonstrated a similar picture of pathogen identification as an observational study by Wang et al., in which immunocompromised patients with systemic corticosteroid exposure were included [21]. A previous study by Gyarmati demonstrated that decreased white blood cell counts among patients with hematologic malignancy were associated with the presence of microbial DNA. In our cohort, eight (35.7%) of 14 neutropenic patients obtain equal or more microbiologic evidence via mNGS, and of which five patients were documented with polymicrobial infection. Neutropenia tends to be less influential with a positive yield rate of mNGS in our cohort. However, polymicrobial infection among patients with febrile neutropenia were considered to be underestimated [22], therefore mNGS might be helpful in obtaining an accurate diagnosis.

Our cohort successfully demonstrated that *P. jirovecii* could be detected from serum samples through mNGS, while previous studies revealed lower diagnostic sensitivity of conventional PCR methods in serum than respiratory specimens [23]. Overall, CMV were detected among seven patients by mNGS; only three patients were confirmed with CMV viremia by conventional viral PCR. Compared with conventional viral assay, mNGS demonstrated it is a faster and more sensitive method in detecting CMV. CMV could be detected by mNGS, even when conventional assay revealed negative results. Previous studies revealed that viral pneumonitis caused considerable mortality among patients with hematologic malignancies or hematopoietic stem cell transplantation [24]. Duan et al. also demonstrated that additional information of viral infection by mNGS predicted a worse outcome, which highlighted the benefit of mNGS [25]. Besides, the diagnosis of viral pneumonitis often relies on bronchoscopy and has been considered underestimated [26]. Blood mNGS tests provide a non-invasive method for pathogen identification, which would bring benefit for those with unexplained sepsis, regardless of broad-spectrum antibiotics administration.

The conventional PCR analysis for detecting pathogens by targeting a specific region of the pathogen genome is limited by the numbers of pathogens in samples (the culture method is also limited by pathogen quantity) and the copy number of targeted genes of the pathogen genome, especially in the early phase of infection. In contrast, over-amplification of PCR causes false positives by an artifact of PCR chimera. On the other hand, the mNGS has increased sensitivity on diagnosis by detecting any part of the genome of pathogens via a shotgun metagenomic profiling tool (i.e., Kraken2). Notably, the sequencing depth of mNGS is a deterministic issue of the detection limit. It is suggested that genome coverage >95% and sequencing depth coverage >100 folds are reliable for the viral detection [27]. Using the shotgun-based high-throughput sequencing technique satisfies the above conditions for detecting rare pathogen fragments in the NGS libraries. We successfully find more CMV-positive cases when the conventional tests are negative.

However, mNGS still showed its limitation in several aspects. First, similar to conventional blood cultures, the yield rates of mNGS could be influenced by the timing of the specimen obtained. A previous study by Grumaz et al. revealed that the rate of positive mNGS results was constant over the different time points after sepsis developed, while the positivity of blood culture decreased at later time points [28]. Camargo et al. indicated that cell-free DNA sequencing could still identify fungus such as *P. jirovecii* or Aspergillus species among patients receiving effective antifungal agents [29]. Another study by Miao et al. also revealed that mNGS is less affected by prior antibiotic exposure, which showed opposite findings to our cohort. In our cohort, mNGS still possessed greater sensitivity for those who had serum sampled prior to effective antimicrobial agent exposure [7]. We inferred that yield rate are still associated with immune status. Research by Wang et al. demonstrated that mNGS showed especially greater sensitivity than conventional methods among those with higher cumulative steroid dose. However, positive yield rate still decreased along the increase of cumulative steroid dose [21].

In addition, the results of mNGS should be interpreted along with clinical symptoms and conventional methods due to possible discrepancies between mNGS and conventional methods. Until now, there is still no strict standard for distinguishing whether a pathogen detected by mNGS is pathogenic or colonization. Since infection focus is often unclear among patients with febrile neutropenia, any positive results of mNGS should be reviewed with caution according to clinical symptoms and signs. Take an example of Case 17, which mNGS revealed HSV-1 and *C. neoformans*; there was no conclusive microbiologic evidence via conventional methods. However, since there was no obvious neurologic or respiratory symptom, only empirical cefepime was prescribed. Fever resolved after neutropenia recovered seven days later. No direct evidence of HSV-1 infection or cryptococcosis was documented during his hospitalization. Nikkari et al. demonstrated that the presence of bacterial DNA in the blood of healthy people might be a result of physiological translocation of bacteria from the oral or gastrointestinal tract, which did not induce sepsis [30,31].

On the other hand, pathogens which are commonly considered as environmental contaminants would probably be omitted by mNGS database. For example, in our cohort, case no. 1 was an 83-year-old AML patient with febrile neutropenia, who was later confirmed with *B. cereus* bacteremia via two sets of blood culture. However, preliminary mNGS data resulted in negative, but with an extra profuse signal (reads > 100) of B. cereus in the session of environmental contaminants. Pathogens that used to be considered as contaminations could result in severe infection, such as Bacillus and coagulase-negative Staphylococcus, which should not be ignored when interpreting mNGS results [32]. As a result, we supported mNGS as an assisting method for pathogens identification; however, conventional methods remained necessary.

In our study, the results of mNGS were not timely available due to high cost and time-consuming data interpretation. Therefore, the impact of mNGS methods on the physicians’ preference of antimicrobial agent administration remained unclear. Besides, the establishment of the mNGS platform might still be a barrier for hospitals to obtain mNGS as a routine diagnostic method, since efforts to standardize and validate the lab processing and data interpretation for timely diagnosis required hardware upgrades and further training of laboratory skills.

## 5. Conclusions

Our study demonstrated the adjunctive role of mNGS in the identification of pathogens in hematologic patients with sepsis. Interpretation of mNGS results should be evaluated along with the patient’s clinical presentations.

## Figures and Tables

**Figure 1 microorganisms-09-02309-f001:**
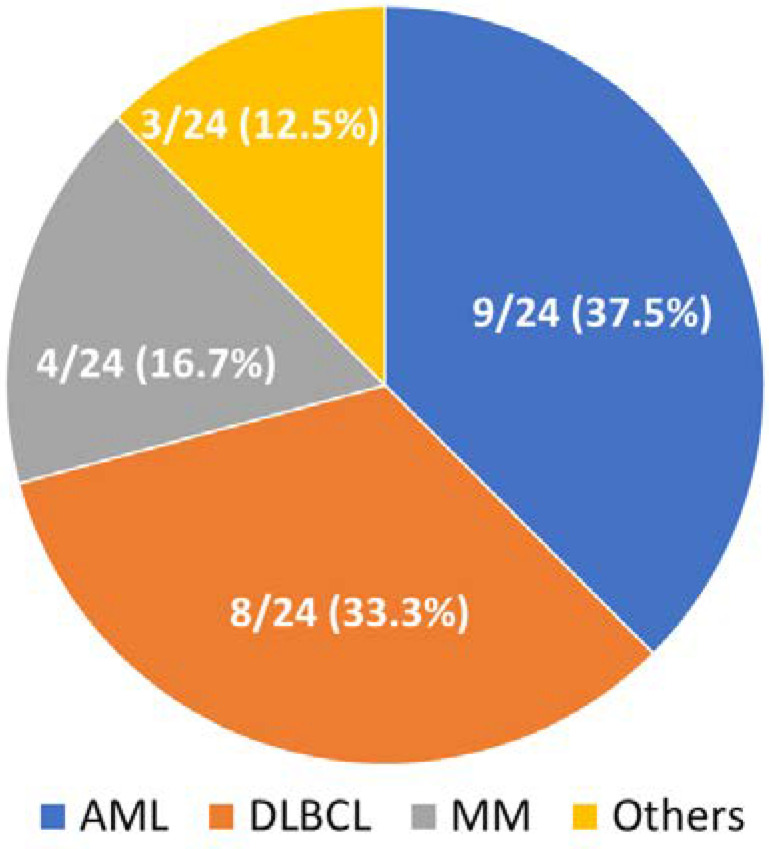
Pie chart demonstrating the underlying hematologic malignancies in the 24 enrolled patients.

**Figure 2 microorganisms-09-02309-f002:**
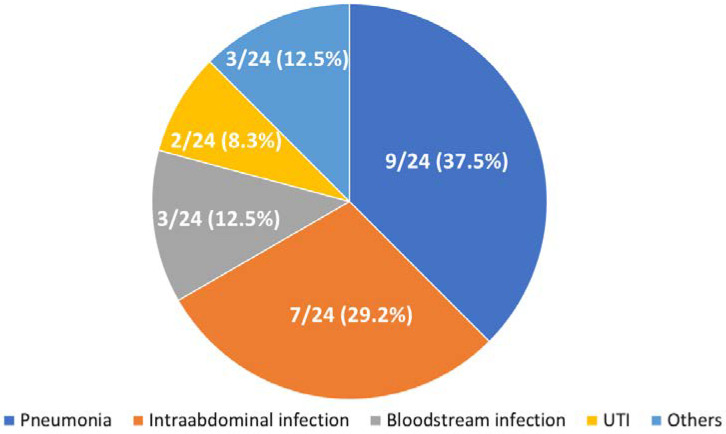
Pie chart demonstrating the infection focus in the 24 enrolled patients.

**Figure 3 microorganisms-09-02309-f003:**
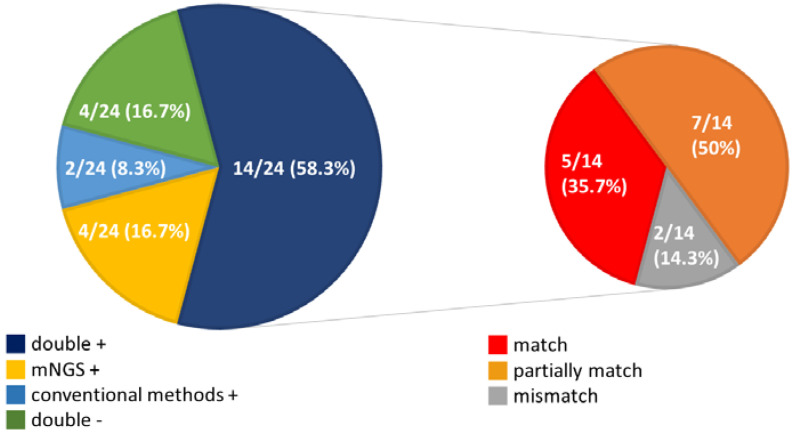
Pie chart demonstrating the positivity distribution of mNGS and conventional methods in the 24 enrolled patients.

**Table 1 microorganisms-09-02309-t001:** Clinical information and microbiologic evidence from conventional method and metagenomic next generation sequencing of the enrolled patients.

No.	Age/Gender	Underlying Diseases	Neutropenia	Clinical Diagnosis of Infection Disease	Blood Cultures	Additional Laboratory Tests	mNGS Results (Reads, Relative Abundance [%])	Interval between Fever Onset and mNGS Sampling (Days)	mNGS Sampled before Effective Antimicrobial Agents
Information of conventional diagnostic method equal to mNGS (*n* = 9)									
1	83/M	AML	Y	BSI	*Bacillus cereus*		*B. cereus* (100, 0.627)	3	Y
2	83/M	AML	Y	CAP	*Pseudomona aeruginosa*		*P. aeruginosa* (4022, 21.86)	1	Y
3	70/M	AML	Y	HAP	*Klebsiella pneumoniae*		*K. pneumoniae* (55, 0.45)	1	N
4	74/M	DLBCL	N	HAP		Serum and sputum Cytomegalovirus PCR (+)	CMV (2233, 10.63)	1	Y
5	66/M	MM	N	HAP		Serum and sputum CMV viral load: (+) Bronchial wash *Pneumocystis jirovecii* PCR (+)	CMV (19, 0.17) *P. jirovecii* (16, 0.14)	1	N
6	73/M	AML	N	IAI				2	-
7	69/M	DLBCL	N	HLH				3	-
8	80/M	AML	Y	HAP				0	-
9	44/M	AML	Y	IAI				1	-
More information of mNGSs (*n* = 10)									
10	64/F	DLBCL	N	UTI	*K. pneumoniae*		*K. pneumoniae* (46, 0.51) CMV (27, 0.3)	1	N
11	75/M	DLBCL	N	CRBSI	*Enterobacter cloacae*		*E. cloacae* (229, 1.93) *Enterococcus faecium* (26, 0.22) *Acinetobacter baumannii* (18, 0.15)	0	Y
12	76/M	AML	Y	VAP	*Burkholderia cepacia complex* *E. faecium*		Human mastadenovirus C (22,355, 79.62) *E. faecium* (47, 0.17) *Bukholderia ubonensis* (13, 0.05)	0	Y
13	61/F	T-LGLL	Y	IAI	*Escherichia coli*		*Shigella dysenteriae* (48, 0.09) *K. pneumoniae* (45, 0.08) *E. cloacae* (24, 0.04) *Candida albicans* (10, 0.02)	0	Y
14	53/M	MM	Y	IAI	*K. pneumoniae*		*K. pneumoniae* (113, 1.66) *P. aeruginosa* (18, 0.26) *E. coli* (5, 0.07)	2	Y
15	76/M	MM	N	UTI	*E. coli*		CMV (5260, 49.51) *E.coli* (33, 0.31) *E. faecium* (31, 0.29)	4	N
16	66/M	MM	Y	IAI			*K. pneumoniae* (210, 2.14) CMV (44, 0.34) *C. albicans* (4, 0.03)	0	Y
17	55/M	AML	N				HSV-1 (42, 0.18) *Cryptococcus neoformans* (4, 0.02)	2	N
18	62/M	DLBCL	N				HHV-6 (5310, 23.58)	3	N
19	61/M	DLBCL	Y	HAP			CMV (14, 0.05)	1	N
More information of conventional methods (*n* = 3)									
20	64/F	DLBCL	N	HAP	*K. pneumoniae*	Serum CMV viral load: (+)	CMV (39, 0.33)	1	N
21	61/M	MCL	Y	IAI	*Aeromonas veronii*			2	N
22	37/M	ALL	Y	HAP	*E. coli*			1	N
Inconsistent results between mNGS and conventional methods (*n* = 2)									
23	82/M	DLBCL	Y	IAI	*S. enterica* *E. faecium*		*C. albicans* (310, 1.86) *S. enterica* (177, 1.06) CMV (46, 0.28)	2	N
24	80/F	AML	Y	BSI	*Candia tropicalis*		*E. coli* (5, 0.05)	2	N

Abbreviation: ALL, acute lymphocytic leukemia; AML, acute myeloid leukemia; BSI, bloodstream infection; CAP, community-acquired pneumonia; CMV, cytomegalovirus; CRBSI, catheter-related bloodstream infection; DLBCL, diffused large B cell lymphoma; HAP, hospital acquired pneumonia; HHV-6, human herpes virus-6; HLH, hemophagocytic lymphohistiocytosis; HSV-1, herpes simplex virus -1; IAI, intra-abdominal infection; MCL, mantle cell lymphoma; MM, multiple myeloma; mNGS, metagenomic next generation sequencing; PCR, polymerase chain reaction; T-LGLL, T-cell large granular lymphocytic leukemia; UTI, urinary tract infection.

**Table 2 microorganisms-09-02309-t002:** Distribution and classification of high-quality reads among the blood specimens.

No.	Raw Reads	Clean Reads	Human Reads	Human (%)	Bacterial Reads	Bacterial (%)	Fungal Reads	Fungal (%)	Viral Reads	Viral (%)	Unclassified Reads	Unclassified (%)
1	61,631,438	372,544	59,667,222	96.8	3874	0.006493	80	0.000134	16	0.000027	368,574	98.934354
2	47,018,822	751,060	45,392,646	96.5	6265	0.013802	549	0.001209	47	0.000104	744,199	99.086491
3	44,622,470	754,142	43,207,034	96.8	2199	0.005089	585	0.001354	32	0.000074	751,326	99.626596
4	61,310,234	1,295,834	58,817,288	95.9	2511	0.004269	889	0.001511	2293	0.003899	1,290,141	99.560669
5	47,892,512	484,488	46,624,618	97.4	1709	0.003665	297	0.000637	37	0.000079	482,445	99.578318
6	140,925,496	1,038,042	136,530,428	96.9	4472	0.003275	504	0.000369	18	0.000013	1,033,048	99.518902
7	48,307,008	370,708	46,691,368	96.7	6860	0.014692	168	0.000360	81	0.000173	363,599	98.082318
8	34,052,126	255,536	33,138,968	97.3	2673	0.008066	93	0.000281	11	0.000033	252,759	98.913265
9	22,177,606	485,240	21,191,972	95.6	33,501	0.158083	325	0.001534	223	0.001052	451,191	92.983060
10	37,286,458	413,648	36,212,734	97.1	1441	0.003979	260	0.000718	89	0.000246	411,858	99.567265
11	89,094,784	506,686	85,454,082	95.9	1976	0.002312	98	0.000115	438	0.000513	504,174	99.504229
12	20,962,736	348,342	20,138,688	96.1	1644	0.008163	268	0.001331	5289	0.026263	341,141	97.932779
13	31,182,134	912,828	29,495,118	94.6	62,320	0.211289	594	0.002014	471	0.001597	849,443	93.056195
14	61,919,982	393,804	58,928,694	95.2	551	0.000935	96	0.000163	1	0.000002	393,156	99.835451
15	20,962,736	348,342	20,138,688	96.1	1644	0.008163	268	0.001331	5289	0.026263	341,141	97.932779
16	35,937,874	418,856	34,893,316	97.1	5419	0.015530	227	0.000651	79	0.000226	413,131	98.633182
17	167,008,212	1,087,152	161,133,136	96.5	4440	0.002755	309	0.000192	61	0.000038	1,082,342	99.557560
18	70,752,128	523,784	68,587,748	96.9	8907	0.012986	227	0.000331	5409	0.007886	509,241	97.223474
19	44,666,980	754,222	42,965,016	96.2	30,655	0.071349	521	0.001213	78	0.000182	722,968	95.856127
20	35,576,312	367,184	34,538,460	97.1	4342	0.012571	236	0.000683	67	0.000194	362,539	98.734967
21	87,638,198	467,968	83,603,160	95.4	2608	0.003119	53	0.000063	15	0.000018	465,292	99.428166
22	81,554,252	421,902	78,379,234	96.1	3041	0.003880	73	0.000093	4	0.000005	418,784	99.260966
23	71,295,070	479,434	68,314,830	95.8	4688	0.006862	477	0.000698	75	0.000110	474,194	98.907045
24	54,129,834	467,204	52,886,676	97.7	736	0.001392	264	0.000499	17	0.000032	466,187	99.782322

The raw reads are trimmed together with the host reads and go through a series of processes (see Materials and Methods), and then become clean reads. The clean reads still included bacterial reads, fungal reads, virus reads, and other unclassified reads.

## Data Availability

Sequencing data have been deposited in the NCBI Sequence Read Archive (SRA) under the accession number PRJNA704313.

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
