# Peer review of "Clinical Application of Metagenomic Next-Generation Sequencing in Patients with Hematologic Malignancies Suffering from Sepsis"

_microorganisms, 2021, doi:10.3390/microorganisms9112309_

Round 1
Reviewer 1 Report
This is another study that illustrate the superiority of NGS approaches to diagnose bacterial and other organism, that are associated with sepsis. Thus the results are confimatory and not new.
With nine patients one cannot conduct a statistical analysis. The results are more or less descriptive. However, the manuscript would greatly improve, if the authors were more clear on the methods especially the bioinformatics paragraph leaves much to be desired. To give two examples: The definition of "clean reads" is unlcear. The clean reads are only a tiny fraction of the raw reads and this is disturbing. I could not reproduce the percentages for the columns Bacteria, Fungal,Virus reads in %. Nor could I tell, when an organism is identified in the NGS data. Here a more detailed description is mandatory.
The discussion is okay, for my taste a bit to extensive given the fact that the results are based on a few patients only. but the points made in the discussion are all valid from a global perspective. However, I disagree with the last paragraph in the discussion. Costs for NGS sequencing are low and once a solid pipeline from patient to bioinformatics interpretation is established in a clinical setting, then the NGS diagnosis is fast compared to standard approaches. The authors should not confuse the development and the pilot application with a routine setting.
Reviewer 2 Report
In this paper, Liu and colleagues have demonstrated that the possible instrumentality of mNGS as pathogens identification tools in hematologic sepsis patients. This manuscript is well written and holds authors’ messages nicely, in which this mNGS approach is able to support estimation of any involvement of specific pathogens in septic progression of those patients. As described also by the authors in the Introduction, some of the key methods including mNGS might have been shared with those which were used in previous study (PMID 26996149), which may count against the authors. The quantitative reinforcement of clinical utilization for further addition of pathogenic identification, and data comparison between mNGS and conventional methods (blood culture, serology test, and virus RT-PCR) might be considered to let it keep a merit of current manuscript. But, the results obtained via these conventional methods were shown only as Table and brief explanations in the text. To my understanding it is crucial to separately include the data obtained from these methods. The authors would need to revise the manuscript by including new Figures (e.g., graphs) along with the descriptions. Also, I have some additional comments, which should be addressed by the authors.
The experimental materials and procedures for methods of blood culture, serology test, and virus RT-PCR are not sufficient (lines 75~80). The authors should describe those conventional methods and materials for more details in separate subsections.
It is recommended to include any schematic illustrations using pie charts and Venn diagrams showing the tendency of sequencing reads distributions using the authors’ current accumulated data. This inclusion would make current manuscript more supportive in efficient comprehension of authors’ findings.
The citation and discussion of recent reports (such as PMID 30110400, 33505547, 33623780, 33893492, etc.) relevant to this study might be missing. The authors may wish to include descriptions with some of those references in Discussions for general readership.
Round 2
Reviewer 2 Report
I have a few comments, which can be quickly addressed.
In the Figures 1-3, data show frequency. The authors should consider to modify current numbers in the pie charts to 9/24 (37.5%), 8/24 (33.3%), etc.
In line 233, I wonder whether the word “100x” is correct.
In lines 293-294, “Acknowledgments” should be revised.